# Scanning Super/Ultrapulsed CO_2_ Laser Efficacy in Laryngeal Malignant Lesions

**DOI:** 10.3390/medicina58020200

**Published:** 2022-01-28

**Authors:** Stefano Dallari, Luca Giannoni, Alessandra Filosa

**Affiliations:** 1Otorhinolaryngology Unit, Department of Surgery, “A. Murri” Hospital—ASUR Marche, Area Vasta n. 4, 63900 Fermo, Italy; 2Reasearch and Developement, El.En. Group, 50041 Florence, Italy; l.giannoni@elen.it; 3Pathology Department, “A. Murri” Hospital—ASUR Marche, Aree Vaste n. 4 and 5, 63900 Fermo, Italy; alessandra.filosa@sanita.marche.it

**Keywords:** transoral laryngeal microsurgery, laser in ENT, laryngeal cancer, scanning laser technology

## Abstract

*Introduction:* The authors review their experience in transoral laryngeal microsurgery (TLM) that they performed with two different CO_2_ laser devices from the same company, which were both equipped with a micromanipulator and digital scanner. *Material and Methods:* A total of 91 glottic and glotto-supraglottic cancers were treated during the years 2009–2016 and then analyzed in relation to the laser performances and the long-term oncologic results. *Results:* Laser devices proved to be very efficient and the UP mode was confirmed to be the best in terms of cutting precision and lowest thermal damage. *Conclusions:* CO_2_ laser TLM is the preferred option for the majority of small–medium size glottic and supraglottic cancers and may also be used for bigger tumors, especially in older patients.

## 1. Introduction

In the early 1970s, Strong and Jako pioneered the so-called laryngeal laser surgery [1].

The CO_2_ laser, targeting water with its 10,600 wavelength, guarantees excellent cutting properties and low thermal spread [2,3]. It started to be used in microsurgery and endoscopic surgery, mainly in the field of otolaryngology and head and neck diseases [4].

Wolfgang Steiner holds the well-deserved title of father of the transoral laryngeal oncologic CO_2_ laser microsurgery [5].

From the 1980s onwards, several authors significantly contributed to this field [6,7,8,9], and the technology and performance of laser systems improved significantly. Micro-spot micromanipulators and electronic scanning units were introduced, thus optimizing and simplifying the surgical techniques and encouraging more and more surgeons to adopt the CO_2_ laser, which became the gold standard in transoral laryngeal microsurgery (TLM) [10].

In 2000, on behalf of the European Laryngological Society (ELS), an ad hoc committee proposed a classification of the cordectomies [11] that was widely adopted. This contributed to the analysis and comparison of the oncologic and functional results of TLM or, as recently proposed [12], CO_2_ TOLMS (CO_2_ transoral laser microsurgery).

This classification was further updated in 2007 [13] with the introduction of a new category of cordectomy, type VI, and in 2009 [14] with a classification for supraglottic procedures.

This paper analyses and compares the experience gained with the SmartXide^®^ HS and SmartXide^2®^ CO_2_ laser systems in a series of 91 laryngeal cancers treated with TLM during the years 2009–2016 (Table 1).

The focus is on the technical characteristic of the CO_2_ laser equipment and the long-term oncologic results.

## 2. Materials and Methods

### 2.1. Laser Equipment

The SmartXide HS CO_2_ laser has a DC-excited source and the emission modes are continuous wave (CW), pulsed wave (PW) and superpulsed (SP) wave. It can have 30 W or 50 W of maximum power.

The SmartXide^2^ system is equipped with an RF-excited CO_2_ laser source of 60 W or 80 W maximum power. The lasers in the Smartxide^2^ family have the so-called DEKA Pulse Shape Design (PDS^®^) technology, which enables great versatility of emissions from the continuous mode to various kinds of pulsed modes, up to the ultrapulsed (UP) mode.

The SmartXide^2^ family can also be equipped with a 50 W 980 nm diode laser source, for otologic surgery and is useful in procedures in which more coagulative power is needed. SmartXide^2^ TRIO (60 W) is the laser that the senior author (S.D.) is currently utilizing. In the TRIO model, the CO_2_ beam can be delivered by both an articulated arm and a hollow fiber. The laser machines previously used were the SmartXide HS and a SmartXide^2^ C80.

Both SmartXide HS and SmartXide^2^ systems were high-performing but the UP mode turned out to be more effective than the SP mode, as it delivers much more energy per pulse to the tissue in the same emission time, thus obtaining a faster and char-free tissue cutting and ablation. Technically speaking, to perform an effective CO_2_ laser surgery, the key parameter is the energy per pulse over the tissue ablation threshold, instead of the peak power (Figure 1). A very high peak power that lasts a few microseconds only, does not produce enough energy to effectively ablate the tissue.

The accessories for CO_2_ laser microsurgery consist of a micro-spot micromanipulator (Easyspot Hybrid) and a surgical scanner (HiScan Surgical), which are connected to each other and coupled to the surgical microscope with an adapter (Figure 2a). These accessories were the same for the two laser systems used.

The articulated arm of the laser delivers the laser beam to the scanner. The scanner is electronically controlled by the laser system and can reproduce different scanning shapes with selectable sizes and emission modes, based on the treatment needs (Figure 2b). The scanned beam enters the micromanipulator’s optical zoom and allows the operator to focus the laser on the same operating microscope’s focal length. Finally, a terminal gimbaled mirror, controlled by a mechanical joystick (Figure 2c), allows the surgeon to position the scanned and focused beam on the target. A microswitch, located on the top of the micromanipulator joystick, enables the surgeon to remotely control all the main scanning functions, such as the rotation and size of the cutting/ablation shapes, the selection of the scan’s on/off mode and the fine centering of the beam in the optical zoom, without taking their eyes off the operating microscope.

Concerning the laser setting parameters, the SP or UP emission modes were selected according to the laser used and the tissue to be cut/ablated. The UP mode is more effective in cutting (see above), so it normally needs a lower mean power setting than the SP mode: 2–4 W for the mucosa, 4–6 W for the muscle and 8–10 W for the cartilage. The dwell time was 1.5 ms, used in repeated or continuous scanning modes.

Cutting was mostly performed by using a straight line as the scanning shape. Final brushing, when needed, was performed by using a hexagonal shape, CW as the emission mode, 10–14 W of power, 0.1 ms dwell time and the repeated scanning mode.

### 2.2. Surgical Classification

The series was classified, in accordance with the ELS classifications, as follows:

Cordectomies:

Type I (subepithelial), type II (subligamental), type III (transmuscular), type IV (total), type V a,b,c,d (extended) and type VI (excision of the anterior commissure).

Enlarged procedures:

Few so-called enlarged procedures (hemilaryngectomies) were also performed and the oncologic results are hereby reported.

### 2.3. Patient Selection, Clinical Work-up and Surgical Technique

TNM staging was the main criterion to select, advise and perform a CO_2_ laser TLM. Tis, T1 and T2 tumors were normally eligible, while T3 cases required more factors to be taken into account and discussed with the patient: endoscopy, imaging, age, performance status and the patient’s expectations. In relation to pre-cancerous lesions and small malignant tumors, both cold instrument or laser excision were suggested due to the good functional (voice) results that the two procedures may equally achieve. Most of the T1 cases were treated with the CO_2_ laser unless the patient had crucial voice necessities. In our opinion, CO_2_ laser TLM and external radiotherapy have theoretically the same oncologic outcomes. The great disadvantages of radiotherapy are practical and organizational factors including the waiting lists generally being longer, often daily journeys to the radiotherapy center being required and the patient being out of work for seven weeks. Concerning the bigger tumors, the indications were sometimes at the limit and the TLM procedure was decided upon because of the factors previously reported (age and performance status). In such cases, patients, their relatives and care givers were thoroughly informed about the pros and cons, the possibility of a relapse and the consequent necessity to proceed to major surgery. Each patient underwent a complete ENT evaluation, including their history, endoscopic examination with voice recording and endoscopic videos. For small lesions with normal cord motility, imaging was not routinely obtained; while in most of the T1 cases and all the T2 and T3 cases, patients had a contrasted CT scan and/or an MRI performed pre-operatively. Since 2015, chromoendoscopic study (SPIES System, Storz Co., Tuttlingen, Germany) was performed during the pre-operative work-up and systematically during the surgery: at the beginning, during the excision in doubtful situations and at the end of the procedure [15]. In those cases with a histologic-confirmed radical excision, standard endoscopic follow-up visits were carried out as follows: every month for three months, then every two months in the first year, every three months in the second and third year, twice or three times a year in the fourth and fifth year, then once a year. When dealing with bigger tumors or if the patient did not stop smoking, the follow-up might have been tighter. Plane chest X-ray (CXR) was routinely required every six months in the first three years, again with variations related to the volume of the initial tumor. When surgical radicality was uncertain, that is with close margins (with a strip of healthy tissue of 1 mm or less all around the lesion), or with margins involved and suspected microscopic foci left behind, the patient was proposed to have an immediate open surgery (if feasible) and/or radiotherapy. However, generally the strategy was watchful waiting and a contrasted CT scheduled in three months’ time, followed by a further microlaryngoscopy (MLS) for control. This strategy was derived from the experience that the authors gained in treating NMSC of the face. In such cases, quite often close margins or even involved margins, when revised, showed no residual malignancy [16,17]. An endoscopic/imaging-driven wait and see policy was thus adopted with the glottic cancers [18]. Survival curves for disease-specific survival were calculated from the date of the diagnosis using the Kaplan-Meier method [19]. Again, the size of the tumors played a major role in such decisions. Experience confirmed that endoscopic control and CT imaging allowed avoidance of the MLS control in nearly 50% of the cases (as reported in the discussion section). Concerning the surgical technique, a standard TLM procedure was applied with a previous, concomitant and final endoscopic exploration and chromoendoscopic (SPIES) study. Every time the surgeon was concerned about the risk of post-operative impairment of the airway due to edema and/or bleeding, a tracheotomy was prepared without opening the trachea but just packing the wound with an iodoform gauze that was removed after 24 h with suturing. This procedure is called a “safety cervicotomy”. In very few cases a true tracheotomy was performed. The specimens were sent for histology, oriented by staining the superior edge with ink to obtain a precise mapping of the lesions; all samples were fixed in 10% neutral buffered formalin, dehydrated through a crescent ethanol series, embedded in paraffin, serially sectioned (4–5 μm thickness) and stained with haematoxylin and eosin (H&E) for examination with a light microscope.

## 3. Results

### 3.1. Laser Performance

In order to test the cutting properties and thermal damage of both laser systems used, histologies belonging to patients operated on with both the SmartXide HS and the SmartXide^2^ were revised, pointing out the (lateral) thermal damage on the various tissues (epithelium, vocal ligament/connective tissue and muscle). The analysis was carried out both on “light” procedures (type I cordectomies) and big excisions (enlarged procedures). As shown in Figure 3 and Figure 4, the thermal damage is minimal and the mean coagulation along the incision line is 20 ± 6 µm for the SP mode and 16 ± 7 µm for the UP mode, with negligible charring effects in specimens treated in the SP mode. The median values were 17 µm for the SP mode and 13 µm for the UP mode.

### 3.2. Clinical Experience

The whole series was analyzed according to the type of surgery performed (I–V cordectomy and enlarged procedures).

For each type of surgery, several items were considered: histology at the first operation with regard to the margins; necessity of a safety cervicotomy or tracheotomy (see discussion, 4.7); histologic results after a repeated (planned) MLS; and the type of surgery performed to treat tumor recurrence/persistence (Table 2).

In Table 3, the oncologic results and the follow-up data are reported. A survival curve of patients during the follow-up period was also evaluated and reported in Figure 5. Oncologic results describe the final outcome, that is how many patients were cured with the first operation, how many required further surgery to be cured and how many were not cured despite multiple surgeries. With regard to the follow-up, the series was divided into two groups: 42 patients were operated on between 2009 and 2011, and 49 from 2012 to 2016. The former group had a ten-year follow-up while the latter had five years of follow-up.

Table 4 and Figure 6 summarize the statistical analysis in terms of overall survival rate (OSR), that is the patients alive without any sign of diseases at the time of this study; Disease-specific survival (DSS), that is the whole series of patients except the ones dead with the specific disease; and disease-free survival (DFS), that accounts for the patients alive after the first operation.

#### 3.2.1. Type I Cordectomy

When performing a type I cordectomy, the senior author routinely injects saline into the Reinke’s space (“hydrodissection”) in order to better expose the lesion, better find and develop the plane of dissection and further preserve the vocal ligament.

A total of 22 patients, all males, underwent CO_2_ TLM.

A total of 21/22 (95%) patients were cured with 5/21 (23%) patients requiring further surgery. Only one patient (4.5%), who underwent an additional laser excision and subsequent subtotal laryngectomy with neck dissection, died from tracheal relapse while remaining free from tumors at the neo-larynx. At the time of this review, 7 patients are alive with more than a ten-year follow-up, 9 with more than a five-year follow-up and 5 died from other causes, in most of the cases from senectus.

The overall survival rate (OSR) was 73%, disease-specific survival (DSS) rate was 95% and disease-free survival rate was 77%.

#### 3.2.2. Type II Cordectomy

When adhesions to the vocal ligament and/or difficulties in dissecting the Reinke’s space were encountered, the senior author preferred to perform a subligamental cordectomy.

There were a total of 10 patients: 9 males and 1 female.

A total of 10/10 (100%) patients were cured. Of these, 4/10 (40%) patients recovered with further surgery. A total of 8 patients were still alive and NED and 2 patients died from other causes.

OSR was 80%, DSS 100% and DFS 60%.

#### 3.2.3. Type III Cordectomy

There were a total of 16 patients, all males.

A total of 16/16 (100%) patients were cured. Of these, 3/16 (18.7%) patients recovered with further surgery. A total of 10/16 patients were alive and NED (6 with more than a ten-year follow-up and 4 with more than a five-year follow-up). A total of 1/16 patient died from neck recurrence after further surgery and 5/16 died from other causes (senectus).

One safety cervicotomy was performed.

OSR was 62%, DSS 100% and DFS 81%.

#### 3.2.4. Type IV Cordectomy

When performing a type IV cordectomy, the senior author always completely removes the internal pericondrium.

There were a total of 7 patients: 6 males and 1 female.

All patients (100%) were locally cured. A total of 2/7 (28%) patients recovered with further surgery. A total of 3/7 patients were alive and NED. A total of 4/7 patients died from senectus or other causes.

Four safety cervicotomies were performed.

OSR was 43%, DSS 100% and DFS 71%.

#### 3.2.5. Type V Cordectomy

There were a total of 18 patients: 17 males, 1 female.

A total of 17/18 (94%) patients were cured. A total of 9 patients (50%) were cured with the cordectomy and 8 patients (45%) were cured with further surgery. One patient, despite further open surgery, was not cured and died from the disease. A total of 13 patients were still alive and NED. Two patients, despite additional surgery, died from other reasons (one from a heart attack, the other from long-term post-operative complications) while they were negative at the neo-larynx level. Two patients died from senectus.

Five safety cervicotomies and one tracheotomy were performed.

OSR was 72%, DSD 94% and DFS 50%.

#### 3.2.6. Type VI Cordectomy

There were a total of 9 patients, all males.

All 9 (100%) patients were locally cured: 6/9 patients with the cordectomy (67%) and 3/9 patients (33%) with further surgery. A total of 6/9 are still alive and NED. One patient died from metastatic lung disease, one of cerebral hemorrhage and one of senectus.

Two safety cervicotomies and Two tracheotomies were performed.

OSR was 67%, DSS 100% and DFS 67%.

#### 3.2.7. Enlarged Procedures

There were a total of 9 patients, all males.

This small group included wide CO_2_ laser excisions involving the supraglottis and partially extended to the glottis. They have been divided into lateral supraglottic/glottic surgery (8 males) and medial anterior supraglottic/glottic surgery (1 male).

All 9 patients were locally cured: 7/9 (78%) patients with the first operation while 2/9 (23%) patients needed a second surgery. Only 1/9 patient died from metastatic disease, while 3/9 patients are still alive with a five-year follow-up.

A total of 5/9 patients (55%) died from other causes.

Two safety cervicotomies and three tracheotomies were performed.

OSR was 33%, DSS 100% and DFS 78%.

## 4. Discussion

### 4.1. Laser Performance

Scanning technology allowed a substantial improvement in TLM quality. By moving the focused spot on the tissue and reproducing the cutting or ablating patterns at a digitally controlled and selectable speed, the laser–tissue interaction is predictable and reproducible, the depth of ablation is determined and the thermal damage can be negligible and always controlled. This is of the utmost importance for surgical margin evaluation and consequent prognostic chances [20]. Scanning-assisted CO_2_ laser surgery is also very versatile. When leaving the same laser typical cutting parameters, with the chosen scanning shape (e.g., line), but changing the scan mode from repeated (with a selectable pause between one scan and the other) to continuous, it is actually possible to switch from a very low thermal ablation, with a fine depth control (excellent in phonosurgery and type I and II cordectomies), to faster and more coagulative cutting (which is preferred in more vascularized bigger cancer surgery). Scanning superpulsed and ultrapulsed CO_2_ laser systems have both shown to be highly effective, with the latter having a better rate of ablation but the same cure rate. The median values of thermal spread with the ultrapulsed system were lower: 13 µm instead of 17 µm obtained with the superpulsed laser. These results are comparable and even better if compared to other similar data reported in the literature [21].

### 4.2. Staging of the Disease

TNM is the first and standard means of classifying these diseases but it seems too generic when choosing the type of laser excision. In fact, a T1 or a T2 tumor can be actually treated with a type I to type IV cordectomy, where the vocal cord motility is maintained and the issue is the size of the tumor, as quite often happens in a verrucous carcinoma.

The senior author tailors the excision based on the pre-operative work-up (videostroboscopy, chromoendoscopy and imaging) and the intraoperative situation (chromoendoscopy, multi-bloc trans-tumor resection according to Steiner [22] and frozen sections).

### 4.3. Choice of the Disease

Even in bulky tumors, a wide excision (type V cordectomy) has been afforded by taking into account several factors such as the type of patient (see below), pre-operative endo-stroboscopy and imaging, intraoperative chromoendoscopic examination, multiple-block-resection of the tumor and frozen sections. Strict follow-up is crucial in these cases: contrasted CT and/or MRI and MLS control after three–four months are planned. This depends on the histologic report of close or involved margins, or even where there is crucial uncertainty from the surgeon. Day-to-day experience shows that this strategy is applied in about 50% of the cases where high-resolution video-endo-chromo-stroboscopy and imaging (CT, MRI and laryngeal ultrasonography) were often demonstrated to be enough to avoid further general anesthesia. As our results report, there is a very high possibility of curing tumor recurrence with further surgery (usually subtotal or total laryngectomy), provided the patient is committed and strictly followed up. TLM CO_2_ laser re-excisions were performed in a minority of cases.

### 4.4. Type of the Patients

When taking a look at the age ranges of patients (Table 1), it seems quite evident that the larger the TLM laryngeal procedure, the older the patient. The senior author’s philosophy is to offer an older patient a chance of treating even quite a big and infiltrative tumor with a type V cordectomy, or a so-called enlarged procedure, because of its less invasive nature and, again, because of the demonstrated high chance of recovering with a further major surgery. Pre-operative imaging is a crucial aspect for the study of cartilage and paraglottic spaces, in addition to the patient’s performance status. Thoroughly counselling the patients, their relatives and care givers is, as always, mandatory too. On the contrary, in the presence of a younger patient with a vocal cord tumor, where a TLM appears less reliable, for example for a suboptimal exposure, an open subtotal surgery is the first option to consider.

### 4.5. Status of the Margins

The prognostic role of the surgical margins is widely accepted [23]. As shown in Table 2, free margins after the cordectomy were obtained in a variable percentage of cases (type I cordectomy 54%; type II 80%; type III 62%; type IV 71%; type V 45%, type VI and enlarged procedures 56%). After revision surgery, residual positivity or proximity (closeness) or invasion of the surgical margins decreases. These data seem to confirm what has been observed for the skin [16,17], while still validating a second microscopic look, in order to detect an early recurrence that can be re-treated with TLM. It should be noted that in Table 2 there is no strict correspondence between the numbers of free, close or involved margins in the first intervention and at the control MLS. Indeed, according to the watchful waiting observation strategy, some patients with close or involved margins at the first intervention showed no signs of recurrence during follow-up and did not undergo a control MLS. In contrast, patients with clear margins at the first surgery, developed suspicious or clear signs of recurrence and underwent a second surgical procedure that may or may not have shown a new malignancy. This is easily explained by the field cancerization theory and the extensive dysplastic condition of the mucosa. Based on the periodic review of the results, our attitude has become more flexible over time. Nowadays a three–four months watchful waiting policy is currently adopted in each case of close or uncertain margins and also when the involvement is minimal. Endoscopic high-resolution follow-up and stroboscopy, when feasible, coupled with contrasted CT and/or MRI, suggest when MLS is to be repeated or when strict follow-up is required. Patients, their relatives and care givers are thoroughly informed and involved.

### 4.6. Anterior Commissure

The available literature has demonstrated that the malignant tumors of the anterior commissure have a worse prognosis [24]. In these cases, the senior author’s attitude is to perform a wide, subpericondrial excision and classify the procedure as a type VI cordectomy. When a vocal cord neoplasm reaches the ipsilateral hemi-commissure, the latter is included in the laser excision as a type V cordectomy. Excising the anterior commissure leads to the formation of synechiae with significant voice worsening. Many remedial procedures have been proposed. The senior author prefers a patch of buccal mucosa sutured as a keel (unpublished data). In his opinion, synechiae and voice impairment, when operating on the anterior commissure, are very frequent and difficult to manage. This consequence, which can rarely be avoided, might further justify an aggressive approach in most cases, in order to prevent oncologic failures.

### 4.7. Safety Cervicotomy

This procedure was primarily suggested as a result of working in a secondary hospital with the ENT doctor on-call. This experience showed that it is well accepted by patients, without negative drawbacks. We thus consider it worth widely performing. When the risk of post-operative hemorrhage is consistent, a tracheotomy was suggested and performed. Protecting the airway may also further allow widening of the transoral laser excision and this aims to give patients, especially older patients, a chance of a cure through a less invasive procedure.

### 4.8. Rescue Procedures

During this study, radiotherapy (either as an adjunct, precautional or curative step, or as the treatment of local tumor recurrence) was suggested only in two cases. The main reasons for this approach were related to the local reality where radiotherapy facilities are less accessible due to the never-ending long waiting list and a seven-week necessity for daily travel to access it. The authors are also certain that an open surgical option, especially when dealing with a subtotal laryngectomy, might ensure a better quality-of-life, saving radiotherapy as a further option.

### 4.9. Global Results

As Table 3 shows, 90/91 (99%) patients were locally cured. Indeed, only one patient died from laryngeal recurrence, while two patients died from locoregional failure and two patients from distant metastasis. In 63/91 (69%) patients, the first TLM operation was successful, while 24 patients needed a second operation. It should be noted that only 12 total laryngectomies were performed. This means that 60% of the patients (55/91) maintained their laryngeal function. Similar results were obtained in the study of Ozturk et al. [25] regarding a large series of cases with good oncological outcomes presented from the perspective of a single surgeon.

### 4.10. Follow-up

A total of 11 patients had follow-up for longer than 10 years and 11 patients for longer than 5 years. The availability of such long periods of observation and the rarity of patients dead from disease (5/91) is further confirmation that laryngeal cancer, when properly cured, rarely recurs. As already mentioned, an adequately long follow-up is crucial to pursuing a conservative approach, while it is possible to cautiously enlarge the indications of TLM in the case of quite extended laryngeal diseases, especially in older patients. Special attention must be paid to the laryngeal (thyroid cartilage) frame. In our experience, two cases had a recurrence on the external surface of the thyroid cartilage, while the endoscopic endolaryngeal examination was negative. In these cases, laryngeal ultrasonography proved to be very useful and the two patients could be recuperated and cured with open surgery.

### 4.11. Statistical Analysis

As plotted in Table 4, the present series values for OSR, DSS and DFS were 61%, 98% and 69%. OSR was influenced by the patient’s age at the time of the first surgery and by the long follow-up interval. The good percentages of DSS for all categories/extensions of surgery mean that the surgical procedures were always adequate for the tumor size and extension. A quite low percentage of DFS was counterbalanced by the possibility of surgical recuperation for the patients, even with maintenance of some laryngeal function. As already mentioned, in fact, only 12 total laryngectomies were performed (13%) and only one patient (1%) died from a local, laryngeal recurrence. Our data appear comparable, especially for the DSS, to what reported by Peretti and Colleagues. in their large series of 595 Tis-T3 glottic cancers [26], despite the presence in our series of quite big tumors and enlarged/supraglottic procedures.

## 5. Conclusions

Based on our results, TLM is an efficacious treatment for laryngeal malignancies, where a wide range of vocal cord and laryngeal tumors can be fully cured. Improvements in pre-operative diagnostic tools, as well as technical instrumentation, have refined and widened the indications. The scanning-assisted CO_2_ laser systems, with their dedicated high-precision focusing accessories, play a crucial role in terms of cutting and ablating procedures. They guarantee minimal thermal damage, high reproducibility and enhanced safety. In addition to allowing wider, bloodless endoscopic excisions with significantly reduced post-operative edema and easier and quicker recovery, these features ensure a better intraoperative understanding of healthy and pathologic tissues and give the post-operative evaluation of surgical margins the utmost prognostic significance. Better excision performance also means less post-operative morbidity, shorter hospitalization and significant cost-savings. Long-term follow-up further validates the curative results, as already stated in a wide range of scientific literature.

## Figures and Tables

**Figure 1 medicina-58-00200-f001:**
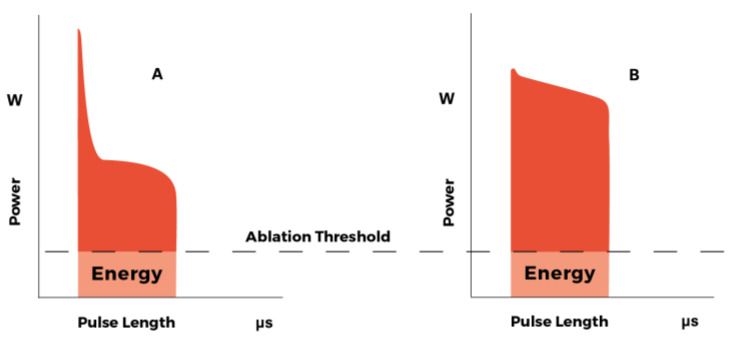
(**A**) DC-(Direct Current) excited CO_2_ laser single pulse (superpulsed emission). (**B**) RF-(radiofrequency) excited CO_2_ laser single pulse (ultrapulsed emission). In RF-excited CO_2_ laser source (**B**), the same pulse length produces more energy over the ablation threshold, compared to the DC-excited one (**A**). (courtesy of DEKA-M.E.L.A., Calenzano, Italy).

**Figure 2 medicina-58-00200-f002:**
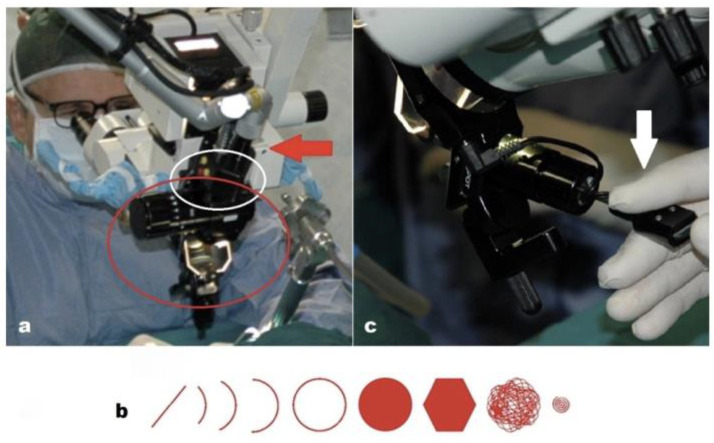
(**a**) Micromanipulator (red ellipse). Surgical scanner (white circle). Connection to the laser arm (red arrow). (**b**) Scanning shapes. (**c**) Joystick (white arrow) with the microswitch on top of the handle.

**Figure 3 medicina-58-00200-f003:**
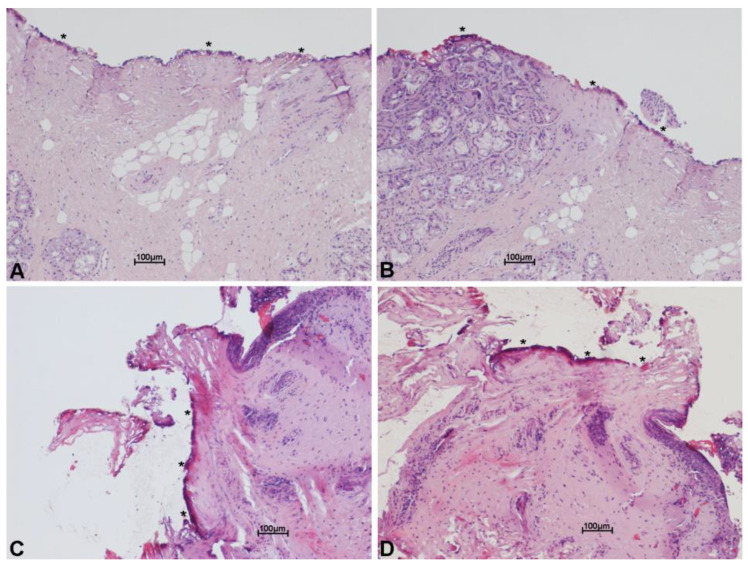
Histological images with hematoxylin and eosin stain, showing minimal thermal effects on the margin of the specimen excised with scanning micromanipulator and superpulsed technology. Thermal damage (*) on the connective tissue (**A**–**D**) and on the glandular epithelium (**B**), along incision line. Scale bars ×10 (**A**–**D**).

**Figure 4 medicina-58-00200-f004:**
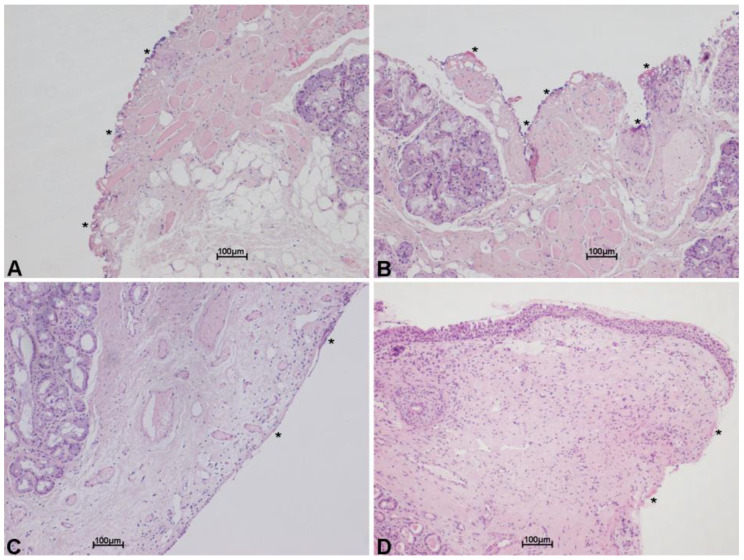
Histological images with hematoxylin and eosin stain, showing minimal thermal effects on the margin of the specimen excised with scanning micromanipulator and ultrapulsed technology. Thermal damage (*) on the connective tissue, along incision line. Scale bars ×10 (**A**–**D**).

**Figure 5 medicina-58-00200-f005:**
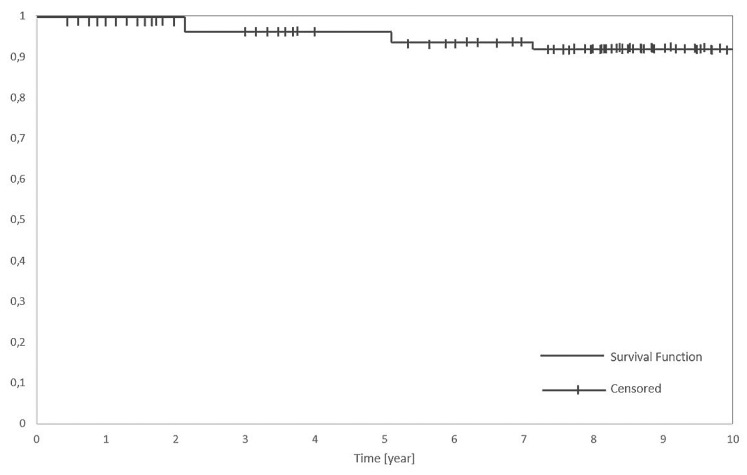
Kaplan-Meier disease-specific survival analysis in a series of 91 patients.

**Figure 6 medicina-58-00200-f006:**
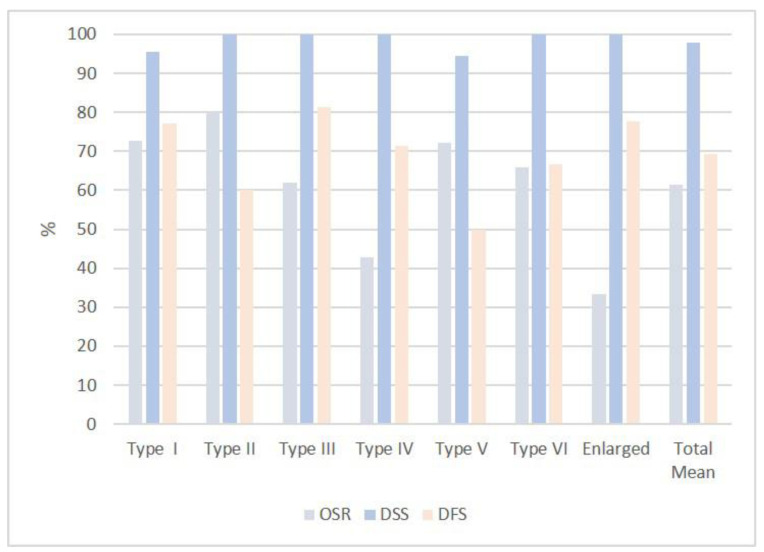
Graphical representation of the percentage of OSR, DSS and DFS rates in all type of cordectomies and enlarged procedures.

**Table 1 medicina-58-00200-t001:** Case series.

	Type I	Type II	Type III	Type IV	Type V	Type VI	Enlarged	Total
Number of patients (2009–2011)	M 11 F -	M 5 F 1	M 6 F -	M 2 F 1	M 6 F1	M 3 F -	M 6 F -	M 39 F 3
Number of patients (2012–2016)	M 11 F -	M 4 F -	M 9 F -	M 4 F -	M 12 F -	M 6 F -	M 3 F -	M 49 F -
Age range								
<50	1	1	-	-	-	1	-	3 (3%)
51–60	8	4	4	1	3	4	-	24 (26%)
61–70	4	1	3	2	6	1	1	18 (20%)
71–80	6	3	5	2	7	2	-	28 (29%)
>80	3	1	3	2	3	1	5	18 (22%)

**Table 2 medicina-58-00200-t002:** Type of operation, histologic results and treatment modalities.

	Type I	Type II	Type III	Type IV	Type V	Type VI	Enlarged	Total
Number of patients	22	10	16	7	18	9	9	91
M F	22 -	9 1	16 -	6 1	17 1	9 -	9 -	88 3
Histologic result after cordectomy								
Margins free	12	8	10	5	8	5	5	53
Margins close	9	1	2		4	1	3	20
Margins involved	1	1	4	2	5	3	1	17
Safety Cervicotomy			1	4	5	2	2	14
Tracheotomy					1	2	3	6
Control MLS (*)								
Margins free	6	3	3		3	2	1	18
Margins close	2		2		1			5
Margins involved	3	1	1		1	1		7
Further surgery to treat recurrence/persistence								
Repeated laser surg.	1	2			2			5
+ subtotal laryng.	5	1	1	1	3	1		12
+ total laryngectomy		1	2	1	4	2	2	12

(*****): See explanations in the text: Section 4.5 Status of the Margins.

**Table 3 medicina-58-00200-t003:** Clinical outcome and follow-up.

	Type I	Type II	Type III	Type IV	Type V	Type VI	Enlarged	Total
Number of patients	22	10	16	7	18	9	9	91
M F	22 -	9 1	16 -	6 1	17 1	9 -	9 -	88 3
Oncologic result after cordectomy								
Locally cured with first surgery	17	6	13	5	9	6	7	63
Locally cured with further surgery	5	4	3	2	8	3	2	27
Locally non-cured (laryngeal failure)					1			1
Follow-up								
10-year follow-up								
Number of patients	11	5	7	3	7	3	6	42
Alive NED (no evidence of disease)	7	3	6	1	6	3		26
Alive with D								-
Dead of D						1 **	1 **	2
Dead of other	4	2	1	2			5	14
5-year follow-up								
Number of patients	11	5	9	4	11	6	3	49
Alive NED	9	5	4	2	7	3	3	33
Alive with D								-
Dead of D	1 *		1 *		1			3
Dead of other	1		4	2	4	2		13

(*) Locoregional recurrence with free larynx (**) metastatic disease.

**Table 4 medicina-58-00200-t004:** Statistical analysis.

	Type I	Type II	Type III	Type IV	Type V	Type VI	Enlarged	Total
Number of patients	22	10	16	7	18	9	9	91
M F	22 -	9 1	16 -	6 1	17 1	9 -	9 -	88 3
Overall survival rate (OSR)	16/22 (73%)	8/10 (80%)	10/16 (62%)	3/7 (43%)	13/18 (72%)	6/9 (67%)	3/9 (33%)	mean (61%)
Disease-specific surv. (DSS)	21/22 (95%)	10/10 (100%)	16/16 (100%)	7/7 (100%)	17/18 (94%)	9/9 (100%)	9/9 (100%)	mean (98%)
Disease-free survival (DFS)	17/22 (77%)	6/10 (60%)	13/16 (81%)	5/7 (71%)	9/18 (50%)	6/9 (67%)	7/9 (78%)	mean (69%)

## Data Availability

The data presented in this study are available on request from the corresponding author.

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
