# Peer review of "Scanning Super/Ultrapulsed CO2 Laser Efficacy in Laryngeal Malignant Lesions"

_medicina, 2022, doi:10.3390/medicina58020200_

Round 1
Reviewer 1 Report
Authors represent their experience in TLM with two different CO2 laser devices, of the same company and both equipped with micromanipulator and digital scanner. There are some modifications needed
1. CO2 laser had long history in treating early glottic cancer, not only for delicate resection of tumor but also fine result for voice function. Please provide pre-op and/or post-op phonation evaluation (such as SPI, jitter, shimmer....etc.) of these two groups of patients
2. I don't really understand the necessity of cervicotomy instead of elective tracheotomy. What's the indication difference in cervicotomy and tracheostomy?
3. The typesetting is irregular for the first and/or second row of each table, please correct.
Reviewer 2 Report
In this report the authors have to underline the data about the TNM patients stage. How many T1? T2? T3? treated by which kind of TLM? What is the policy about the margins (superficial, superficials or deep)? Do you perform as excisional biopsy? or a previuos incisional biopsy? in the 2.2. authors described a Oncologic Classification but it is a surgical classification. The authors spoke about margins free close or positive in table two but it's not clear. When do you consider a positive margin, close or positive? It is mandatory to correlate all the data
Round 2
Reviewer 1 Report
Author had well revised the article and answered the questions.